# Responders to Exercise Therapy in Patients with Osteoarthritis of the Hip: A Systematic Review and Meta-Analysis

**DOI:** 10.3390/ijerph17207380

**Published:** 2020-10-10

**Authors:** Carolien H. Teirlinck, Arianne P. Verhagen, Elja A.E. Reijneveld, Jos Runhaar, Marienke van Middelkoop, Leontien M. van Ravesteyn, Lotte Hermsen, Ingrid B. de Groot, Sita M.A. Bierma-Zeinstra

**Affiliations:** 1Department of General Practice, Erasmus MC University Medical Center Rotterdam, 3000 CA Rotterdam, The Netherlands; Arianne.Verhagen@uts.edu.au (A.P.V.); eljavendel@hotmail.com (E.A.E.R.); j.runhaar@erasmusmc.nl (J.R.); m.vanmiddelkoop@erasmusmc.nl (M.v.M.); s.bierma-zeinstra@erasmusmc.nl (S.M.A.B.-Z.); 2Discipline of Physiotherapy, Graduate School of Health, University of Technology Sydney, Ultimo, NSW 2007, Australia; l.vanravesteyn@gmail.com; 3National Health Care Institute, 1110 AH Diemen, The Netherlands; LHermsen@zinl.nl (L.H.); IGroot@zinl.nl (I.B.d.G.); 4Department of Orthopedics, Erasmus MC University Medical Center Rotterdam, 3000 CA Rotterdam, The Netherlands

**Keywords:** hip osteoarthritis, exercise therapy, responders, meta-analysis

## Abstract

The Outcome Measures in Rheumatology workgroup (OMERACT), together with the Osteoarthritis Research Society International (OARSI) developed the OMERACT-OARSI responder criteria. These criteria are used to determine if a patient with osteoarthritis (OA) ‘responds’ to therapy, meaning experiences a clinically relevant effect of therapy. Recently, more clinical OA trials report on this outcome and most OA trials have data to calculate the number of responders according to these criteria. A systematic review and meta-analysis were performed on the response to exercise therapy, compared to no or minimal intervention in patients with hip OA using the OMERACT-OARSI responder criteria. The literature was searched for relevant randomized trials. If a trial fit the inclusion criteria, but number of responders was not reported, the first author was contacted. This way the numbers of responders of 14 trials were collected and a meta-analysis on short term (directly after treatment, 12 trials *n* = 1178) and long term (6–8 months after treatment, six trials *n* = 519) outcomes was performed. At short term, the risk difference (RD) was 0.14 (95% confidence interval (CI) 0.06–0.22) and number needed to treat (NNT) 7.1 (95% CI 4.5–17); at long term RD was 0.14 (95% CI 0.07–0.20) and NNT 7.1 (95% CI 5.0–14.3). Quality of evidence was moderate for the short term and high for the long term. In conclusion, 14% more hip OA patients responded to exercise therapy than to no therapy.

## 1. Introduction

Hip osteoarthritis (OA) is common cause of disability. In the Netherlands 18.8 out of 1000 men and 33.3 out of 1000 women suffer from hip OA (prevalence in 2018). Osteoarthritis is the tenth cause of Disability Adjusted Life Years (DALY’s) in the Netherlands. The economic cost of osteoarthritis in 2017 in the Netherlands was 433.4 million euros, which is 1.4% of total health cost, of which hip OA is the second cause (knee OA is the first) [1].

Exercise therapy is an important part of the conservative treatment of patients with hip OA [2,3]. Multiple trials have been conducted to study the effect of exercise therapy in hip OA and most of them have shown a positive effect of exercise therapy compared to a non-exercise treatment or no intervention [4,5]. In most of these trials, pain, function, or patient global assessment were the primary outcomes. Combining trials for meta-analysis can often be challenging because of the use of different outcomes and definitions of treatment success. 

An Outcome Measures in Rheumatology (OMERACT) workgroup have put effort into synchronizing future trial outcomes by defining core outcome sets (COS) [6]. After an extensive procedure of discussion and polling, they stated that clinical trials in knee, hip, and hand OA should measure at least four domains: pain, disability, patient global assessment, and for long-term trials, also joint imaging. This was updated in 2019 by adding quality of life and adverse effects to the COS [7]. Subsequently, this OMERACT workgroup together with the Osteoarthritis Research Society International (OARSI) formulated a set of responder criteria [8]. These criteria combine three of the core outcomes: pain, disability, and patient global assessment, in one outcome to define response or non-response to treatment. Some trials already reported on this combined outcome, concerning the effect of exercise therapy in patients with hip OA [9]. Though, trials not reporting this outcome did often measure pain, disability, and patient global assessment separately, and are therefore able to calculate this outcome in their existing dataset. 

In 2016, our department was asked by the National Health Care Institute of the Netherlands to update the existing evidence of three Cochrane reviews on the effect of exercise therapy, compared to no or minimal intervention in patients with hip and knee OA [4,5,10]. The Minister of Health, Welfare, and Sports wanted to evaluate if exercise therapy for patients with hip and knee OA should be reimbursed by the basic health insurance in the Netherlands. Therefore, we conducted a systematic review on the effect of exercise therapy in hip OA, compared to no/minimal intervention. Fifteen studies were included, and four studies reported on responders, although only one used the OMERACT-OARSI set of responder criteria [11]. 

Although this systematic review and meta-analysis showed moderate quality evidence in the short term and high quality evidence in the long term for an effect of exercise therapy compared to no or minimal intervention on pain and function in hip OA, we were interested in this combined outcome of response to treatment. Evaluating the effect of a therapy by looking at number of responders is a very intuitive way. It is closer to practice, since it does not give an average change in a group of patients but a binary outcome per patient and therefore results are easier to interpret for patients and providers of care than continuous outcomes [12]. By using the OMERACT-OARSI responders criteria, we are able to combine data of different studies in an uniform and well-grounded manner [8].Therefore, the aim of this study was to evaluate the existing evidence of the effect of exercise therapy in patients with hip OA, when ‘effect’ is formulated as in the OMERACT-OARSI set of responder criteria. 

## 2. Materials and Methods 

We performed a secondary analysis on the data of the original systematic review (described below).

### 2.1. Search Strategy

We updated the two Cochrane reviews on the effect of exercise therapy in patients with hip OA [4,5]. Therefore, we performed a literature search with the same search terms from these reviews from the date of their last search until March 2019. Main search terms (and derivatives) were osteoarthritis, hip (joint), knee (joint), exercise, sport, physical therapy, rehabilitation, and randomized controlled trials. Literature sources were Cochrane Central Register of Controlled Trials (CENTRAL), MEDLINE, Embase, Cumulative Index to Nursing and Allied Health Literature (CINAHL), PEDro (Physiotherapy Evidence Database), and Web of Science. Detailed information can be found in Supplementary Material S1.

### 2.2. Study Selection

Randomized trials were selected if they fulfilled the following criteria: patients were >18 years old with clinical and/or radiological hip osteoarthritis, the intervention was an active form of exercise therapy under supervision of a (physical) therapist, the intervention was not part of a multidisciplinary or multimodal program and was evaluated as a standalone intervention, the intervention in the control group was usual care (e.g., medication and/or education), and no treatment or waiting list. Studies with control interventions as hot packs, transcutaneous electrical nerve stimulations, and ultrasound were excluded. Furthermore, for this analysis, the outcomes enable us to calculate responders using the OMERACT-OARSI criteria at short term (directly after end of treatment) and/or at long term (6–8 months after end of treatment). 

### 2.3. Risk of Bias Assessment

Risk of bias was assessed with the Cochrane risk of bias tool [13]. This tool has seven domains; random sequence generation, allocation concealment, blinding of participants and personnel, blinding of outcome assessment, incomplete outcome data, selective outcome reporting, and other bias. Each domain is assigned a low, high, or unclear risk of bias. We also assigned studies an overall risk of bias. If a study used a random sequence generation, correct allocation concealment, and intention-to-treat-analysis, it was considered as low risk of bias. A high risk of bias was assigned if less than three domains had a low risk of bias. All other studies were considered to have a moderate risk of bias.

### 2.4. Data Collection

We extracted data on study characteristics: patient population, type of intervention, and type of control group. We also extracted data on results for multiple outcomes concerning responders. If no data was presented on number of responders according to the OMERACT-OARSI criteria, the corresponding author was contacted. We asked the authors whether they were able and willing to calculate the number of responders in the intervention group and control group, or whether they would be willing to provide us with the data to enable us to calculate these numbers.

Selection of studies, risk of bias assessment and data extraction was done by two review authors (A.P.V., C.H.T., E.A.E.R., L.M.v.R., or M.v.M.) independently. In case of disagreement and if no consensus could be reached, a third review author (A.P.V. or M.v.M.) made the final decision.

### 2.5. Outcome

The OMERACT-OARSI set of responder criteria uses pain, function, and patient global assessment to define response to therapy. Response is defined as an improvement in pain or in function ≥ 50% and absolute change ≥ 20, or improvement in at least two of the three following: pain ≥ 20% and absolute change ≥ 10; function ≥ 20% and absolute change ≥ 10; and patient’s global assessment ≥ 20% and absolute change ≥ 10 [8]. Since not all studies collected an outcome of patient global assessment, we agreed to calculate the number of responders with only pain and function in only these studies. This meant that if a patient does not qualify as a responder based on pain and function data, we considered this participant a non-responder. Theoretically, this participant could be a responder if global assessment data would have been available. For outcomes using a Likert scale [14], we converted it to a 0–100 scale. This allowed us to calculate an absolute and relative change from baseline. If necessary, scales were inverted, ensuring that higher scores corresponded with more severe symptoms. If data on pain and/or function were missing, we considered this participant a non-responder.

### 2.6. Analysis

A meta-analysis was performed using a random-effect model in Review Manager 5.3 for short and long term. A risk difference (RD), number needed to treat and 95% confidence interval (95% CI) were calculated. 

Finally, the quality of evidence was determined using the GRADE(Grading of Recommendations, Assessment, Development and Evaluations) approach [15]. Quality was considered high and was subsequently lowered to moderate, low, or very low if one or more of these criteria applied: (a) study limitations: >25% of patients are from studies with an overall high risk of bias; (b) inconsistency: statistical heterogeneity I^2^ > 40% or <75% of patients show results in the same direction; (c) indirectness: results are not suitable to extrapolate to the target population according to expert authors (J.R. and S.M.A.B.-Z.); (d) imprecision: <400 patients; or (e) other: like publication bias or ‘fatal flaw’ (for example selective loss of follow-up).

Sensitivity analysis was done by excluding participants with missing data from analysis (complete cases) and by excluding studies in which no global assessment outcome was available. 

## 3. Results

### 3.1. Selection of Studies

Originally, we performed a review about the effect of exercise therapy in hip and knee OA, therefore, the flowchart shows the references of knee OA during the beginning of the selection process, see Figure 1. Nine out of 12 studies on hip OA from the Cochrane reviews were included; three studies were excluded because they did not fulfill our criteria. Our additional literature search resulted in an extra eight studies. In total 17 studies were potentially eligible for the current analysis. Of these 17 studies, only one study presented results of responders according to OMERACT-OARSI responder criteria. We contacted the authors of the other 16 studies of which 13 responded and were willing to provide us the data on responders. Finally, for this analysis we used the data from 14 studies in total, including a total of 1242 participants.

### 3.2. Study Characteristics

The characteristics of these 14 studies are presented in Table 1. All studies included patients with symptoms (clinical hip OA with or without signs of radiological OA) and most studies (12 out of 14) used the ACR (American College of Radiology) criteria (clinical and/or radiological) to include patients. Group size varied from 14 to 102 patients. Interventions were mostly exercises on land; only one used aquatic exercises. Seven interventions were group-based, five individually-based, and two studies did not specify this. Interventions lasted from 5–12 weeks. Control groups existed of education, medication, waiting list, GP(general practice care), or usual care. All studies measured pain and function, only five studies also measured global assessment.

### 3.3. Risk of Bias Assessment

Ten studies were considered to have a low risk of bias, one study a moderate risk of bias, and three studies a high risk of bias, see Table 2. Blinding participants and outcome assessment were not possible because of the type of intervention. Therefore, almost all studies were considered to have a high risk of bias on the blinding items. Only one study scored low risk of bias on blinding, as they reported that participants had no preference for the treatment or control group [28].

### 3.4. Meta-Analysis

The meta-analysis showed more responders in the exercise group than in the control group, at short term (12 trials, *n* = 1178) and long term (six trials, *n* = 519), see Figure 2. At short term the percentage of responders was 30% in the exercise group and 16% in the control group (RD = 0.14, 95% CI 0.06–0.22, number needed to treat 7.1, 95% CI 4.5–17). At long term the percentage of responders was 26% in the exercise group and 13% in the control group (RD = 0.14, 95% CI 0.07–0.20, number needed to treat 7.1, 95% CI 5.0–14.3). The quality of the evidence for short term outcome was moderate (downgrading because of inconsistency) and high for long term outcome (no downgrading).

### 3.5. Sensitivity Analysis

Complete cases: 95 participants were excluded from the original analysis on short term and 100 participants on long term. Overall risk differences and quality of evidence did not change (see Appendix A). 

Global assessment: In this analysis we only included trials that measured patient global assessment, therefore, the number of responders were calculated according to the full set of OMERACT-OARSI criteria. Only four studies could be included in the meta-analysis on short term (474 participants in total) and three studies for long term (350 participants in total). Risk difference on short term was higher than in the original analysis, although this difference between the two analyses was not statistically significant: RD = 0.20 (95% CI 0.12–0.27, number needed to treat 5.0) and quality of evidence was high (no downgrading). On long term, risk difference stayed the same: RD = 0.13 (0.04–0.21, number needed to treat 7.7), but quality of evidence was moderate because of imprecision (participants < 400), see Appendix A. 

## 4. Discussion

### 4.1. Main Findings

According to the OMERACT-OARSI responder criteria, more patients respond to exercise therapy then to no treatment, usual care, medication, or education only at short term (moderate quality of evidence) and long term (high quality of evidence). The risk difference was 14%, meaning about seven patients should receive exercise therapy to gain one extra responder.

### 4.2. Comparison with Literature

To our knowledge, this is the first meta-analysis on responders to exercise therapy in patients with hip OA. In the Cochrane review on land-based exercise for hip OA, the meta-analysis showed reduction of pain and physical disability and the authors reported a number needed to treat for an additional beneficial outcome (NNTB) of six for both outcomes. This is in line with our results, which is not very surprising since we used the trials of this Cochrane review in our analysis. However, we included more trials and used a different outcome measure. Compared to numbers needed to treat (NNT) published for pharmacological interventions in hip OA, an NNT of 7.1 is quite high. For example, a meta-analysis on steroid injections in the hip in patients with hip OA reported an NNT of 2.4 (95% CI: 1.7–4.2) at 8 weeks post-intervention using the OMERACT-OARSI responder criteria (two studies, *n* = 90) [29]. Another study that pooled two trials on the effect of etoricoxib and celecoxib compared to placebo in patients with knee or hip OA showed an NNT of 4.3 and 4.9, respectively, on the outcome of the OMERACT-OARSI responder criteria [30]. Although if exercise therapy is compared to pharmacologic treatments, it should be noted that the occurrence of adverse effects is low in exercise therapy [5]. Therefore, our results are in line with earlier found effects of exercise therapy and with the considerations of existing guidelines to recommend exercise therapy as non-pharmacological treatment for patients with hip OA [2,3]. After the report of our original systematic review, the National Health Care Institute of the Netherland, recommended the Minister of Health, Welfare, and Sports to reimburse exercise therapy for patients with hip OA [11].

### 4.3. Strengths and Limitations

Although the reporting of OMERACT-OARSI responders is not so common yet, the data was available in most trials and because almost all authors were willing to provide us with these data, we were able to perform a meta-analysis, which has not been done before. 

We followed the same methodological steps as used in the Cochrane reviews on exercise (land-based and water-based) in hip and knee OA, which are considered reliable and evidence-based methods. We aimed to receive a robust answer to the question of whether exercise therapy is effective in hip OA, by combining the data of trials in one uniform way. However, using the OMERACT-OARSI responder criteria in our analysis has some limitations as well. Firstly, the outcomes pain, function, and global assessment were measured on different scales or questionnaires. Especially differences in scale can influence the likelihood of a participant being a responder, or not, in different trials. For example, the WOMAC-pain subscale is measured from 0 (no pain) to 100 (the most pain thinkable) while the HOOS-pain subscale is measured from 100 (no pain) to 0 (the most pain thinkable). This means that participants with less pain at baseline have a higher chance of being responder if pain is measured with WOMAC than with HOOS. To clarify this further: if a participant has a pain score of 30 (scale 0–100) measured with WOMAC questionnaire at baseline, he would qualify as a responder (if for simplicity we only look at pain in the first part of the criteria; a change in pain score of ≥50% and absolute change ≥20), if his pain would decrease with at least 20 points. This participant would have a similar pain score with the HOOS questionnaire around 70 (scale 100–0) and would only qualify as a responder if his pain score would increase (less pain) with at least 35 points. In our analysis we therefore chose to invert scales that were measured 100–0, to ensure uniformity. Another example is the use of Likert-scales. These scales mostly have limited score-options, like 5 or 7. These scales had to be converted to a 0–100 scale to be able to calculate the absolute and relative change, used in the criteria. Nevertheless, then still only 5–7 possible scores are possible, which makes the changes of being responder different than on a more continuous scale, like HOOS or WOMAC. Since we contacted the authors ourselves, we were able to discuss which scales were available in the data and we could uniform this as much as possible. If more trials report on responders and if these data are subsequently combined in meta-analyses, reviewers have to be aware of these differences in measurement. Moreover, authors should be aware of this, by reporting which pain, function, and global assessment outcomes were used and how this was used in the calculation of number of responders.

Secondly, the previous example also shows that there is a possible ceiling effect. One of the authors that provided the data to analyze the responders, also did a brief analysis and found that 46 subjects out of 139 (33% of the population) could not respond to exercise according to the OMERACT-OARSI criteria because of their light symptoms (data not published, with permission of the author [24]). This was also noted by the developers of the responder criteria. They suggested a minimal level of symptoms at entry, if the criteria are used, although no cut-off is determined yet [8].

Thirdly, only a few trials had a measurement on global assessment. We chose to also use the data of the trials without global assessment. By doing so, we could have underestimated the number of responders, because some patients who did not qualify as a responder based on pain and function, could qualify as responder if patient global assessment data was available. Our sensitivity analysis also showed a possible larger effect (quality of evidence was moderate because of the total sample size and the difference with the original analysis not statistically significant) if only trials with a measurement on global assessment were included. 

### 4.4. Implications for Clinical Practice and Research

A priori, we did not specify a minimum for a clinically relevant difference for the risk difference and/or NNT of responders with hip OA. In the literature, we could not find a recommendation or consensus about this. One article on knee OA patients treated with doxycycline mentioned a minimal relevant difference of 20% (NNT five) for the OMERACT-OARSI responder criteria in their study [31]. In comparison with this cut-off and the earlier mentioned pharmacological studies, our effect of exercise therapy on number of responders seems relatively small. This means for clinical practice, that we should realize that a large group of patients will not respond to exercise therapy and it therefore seems important for clinicians to be able to predict which patients are more likely to benefit from exercise therapy. One study evaluated predictive factors for being a OMERACT-OARSI responder to exercise therapy and found that hip OA patients with unilateral hip pain, age of ≤58 years, pain of ≥6/10 on a numeric pain rating scale, 40-m self-paced walk test time of ≤25.9 s, and duration of symptoms of ≤1 year were predictive for response to exercise therapy [32]. In addition, currently a research collaboration is ongoing to study effect modifiers for exercise therapy in knee and hip OA with individual participant data meta-analyses in a databank of trials (OA trial bank) [33]. Lastly, more consensus about the use of scales, measurements, and population (e.g., severity of symptoms) within these criteria, could improve uniformity and comparability of this measurement.

## 5. Conclusions

There was moderate quality evidence in the short term (directly after treatment) and high quality evidence in the long term (6–8 months after treatment) that exercise therapy is effective in patients with hip OA, when compared to no or minimal intervention, considering the OMERACT-OARSI responder criteria, although the magnitude of this effect seems relatively small.

## Figures and Tables

**Figure 1 ijerph-17-07380-f001:**
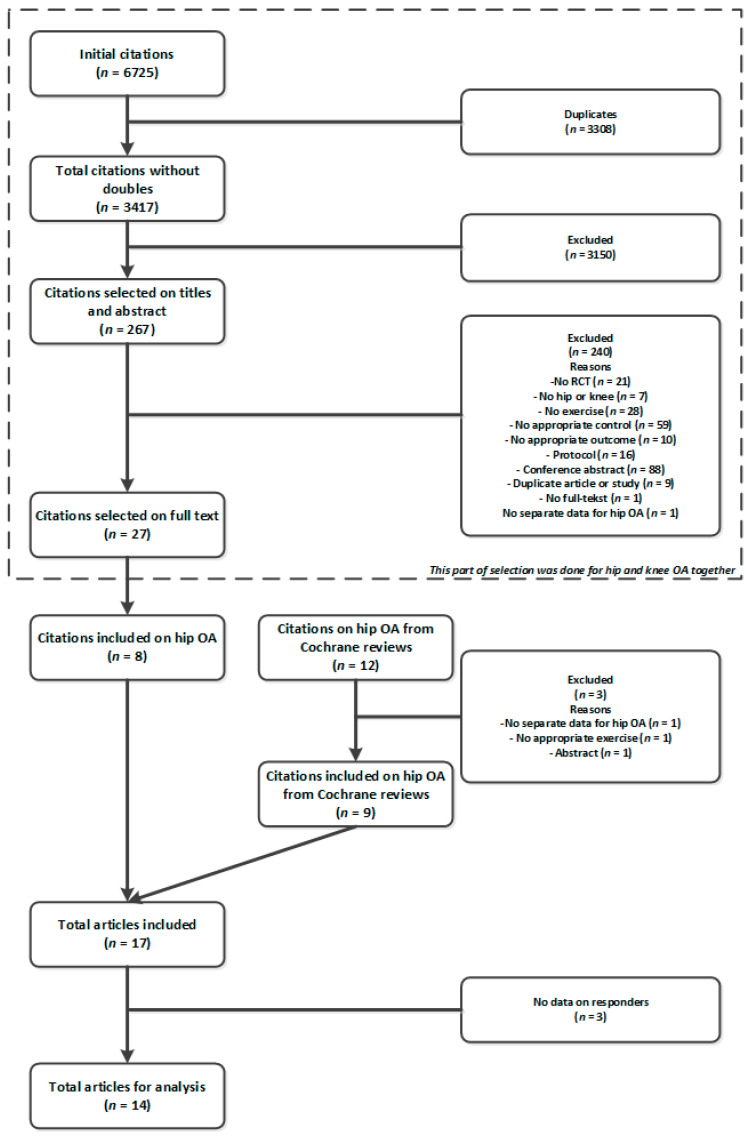
Flowchart selection of studies. OA = osteoarthritis, RCT = randomized controlled trial.

**Figure 2 ijerph-17-07380-f002:**
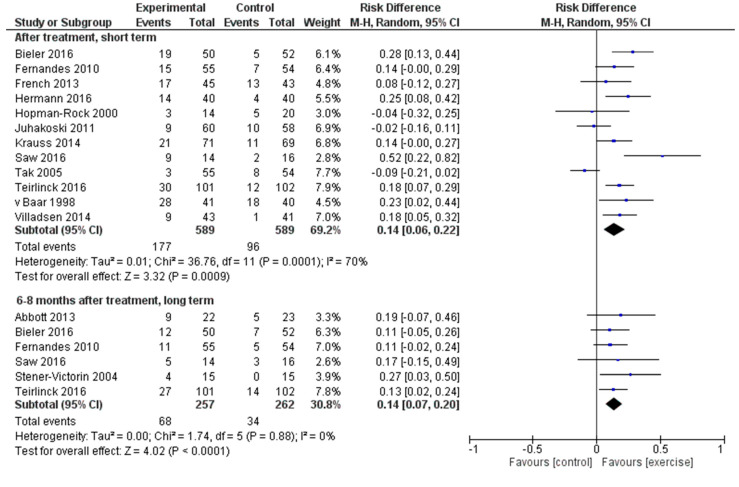
Meta-analysis, forest plot. 95% CI = confidence interval; M-H = Mantel-Haenszel test.

**Table 1 ijerph-17-07380-t001:** Study characteristics.

Study	Population	Intervention	Control	Measurements	Overall Risk of Bias ^#^
V Baar 1998 [16] *N* = 81	Clinical ACR	Exercise (*n* = 35). Individual physiotherapy program (12 weeks, 1–3×/week, 30-min sessions) + GP education + medication if necessary.	GP education + medication if necessary (*n* = 33).	*After treatment:* Function: IRGL Pain: VAS pain past week Global assessment: global perceived effect 8-point Likert scale	Low
Hopman-Rock 2000 [17] *N* = 34	ACR	Exercise (*n* = 14). Group sessions (6 weeks, 1×/week, 60-min classes) + 1×/week education.	Waiting list (*n* = 20).	*After treatment and 6 months after treatment:* Function: IRGL-mobility Pain: VAS pain last month Global assessment: -	High
Stener-Victorin 2004 [18] *N* = 45	Radiological ACR Patients on waiting list for hip replacement	Aquatic exercise (*n* = 15). Group session (5 weeks, 2×/week, 30 min) + education. Two group meetings lasting 2 h each concerning hip anatomy, disease process, and advice on physical activities.	Education (*n* = 15). Two group meetings lasting 2 h each concerning hip anatomy, disease process, and advice on physical activities.	*After treatment and 6 months after treatment:* Function: Disability Rating Index Pain: VAS pain during motion Global assessment: -	High
Tak 2005 [19] *N* = 109	Clinical ACR	Exercise (*n* = 55). Group session (8 weeks, 1×/week strengthening + home program, 60-min) + education.	GP care (*n* = 54).	*After treatment and 3 months after treatment:* Function: Sickness Impact Profile – mobility Pain: VAS pain last month Global assessment: -	High
Fernandes 2010 [20] *N* = 109	Radiological ACR and symptoms (Harris Hip Score 60–95)	Exercise (*n* = 55). Individually based (12 weeks, 2×/week) + patient education.	Patient education (*n* = 54).	*After treatment and 6 months after treatment:* Function: WOMAC Pain: WOMAC Global assessment: -	Low
Juhakoski 2011 [21] *N* = 120	Radiological and clinical ACR, K–L grade >1.	Exercise (*n* = 60). Group sessions (12 weeks,1×/week, 45 min, + 4 booster sessions 1 year later) + GP care	GP care (*n* = 60).	*After treatment:* Function: WOMAC Pain: WOMAC Global assessment: -	Low
French 2013 [22] *N* = 88	Radiological and clinical ACR	Exercise (*n* = 45). Individually provided ‘standardized’ exercise program (8 weeks, 6–8 sessions, 30-min) + daily home exercise program (aerobic walking/cycling/swimming 30 min)	Waiting list (*n* = 43).	*After treatment:* Function: WOMAC-PF Pain: NRS Global assessment: global rating of change 7-point Likert scale	Low
Abbott 2013 [9] *N* = 45	Clinical ACR	Exercise (*n* = 22). Individually provided by physiotherapist, 50 min (9 weeks, 7 sessions + 2 booster sessions week 16).	GP care (*n* = 23).	*8 months after treatment:* Function: WOMAC Pain: WOMAC Global assessment: global rating of change	Low
Villadsen 2014 [23] *N* = 84	Scheduled for hip replacement because of symptomatic OA	Exercise (*n* = 43). Neuromuscular training (8 weeks, 2×/week, 60 min) + education (written information, also on various exercises)	Education (*n* = 41). Written information, also on various exercises	*After treatment:* Function: HOOS Pain: HOOS Global assessment: -	Low
Krauss 2014 [24] *N* = 140	Clinical ACR	Exercise (*n* = 71). Group sessions (12 weeks, 1×/week, 60–90 min, 2×/week home exercises, 30–40 min).	Control (*n* = 69). No intervention	*After treatment:* Function: WOMAC Pain: WOMAC Global assessment: -	Low
Teirlinck 2016 [25] *N* = 203	Clinical ACR	Exercise (*n* = 101). Individual therapy (12 weeks, 12 sessions, 3 booster sessions in 5th, 7th, and 9th month) + GP care.	GP care (*n* = 102).	*After treatment and 6 months after treatment:* Function: HOOS Pain: HOOS Global assessment: Recovery 7-point Likert scale	Low
Hermann 2016 [26] *N* = 80	Scheduled for hip replacement	Exercise (*n* = 40). Pre-operative progressive explosive resistance training (10 weeks, 2×/week, 60 min).	Usual care (*n* = 40).	*After treatment:* Function: HOOS Pain: HOOS Global assessment: -	Low
Saw 2016 [27] *N* = 30	Waiting list for hip replacement, radiological and clinical ACR	Exercise (*n* = 14). Group sessions by physiotherapist (6 weeks, 1×/week, 120 min) + education.	Usual care (*n* = 16).	After treatment and 6 months after treatment: Disability: Health Assessment Questionnaire - functional disability index Pain: Brief Pain Inventory Global assessment: -	Moderate
Bieler 2016 [28] *N* = 152	Clinical ACR, age >60	Exercise (*n* = 50). Group sessions, strengthening/resistance exercises (16 weeks, 3×/week, 60 min).	Counseling + education (*n* = 52).	*After treatment and 8 months after treatment* Function: WOMAC Pain: WOMAC Global assessment: 5-point Likert scale	Low

^#^ Low Risk of Bias (RoB): randomization appropriate + concealed + ITT analysis; high RoB: <3 items low risk; moderate RoB: all else. Abbreviations: GP = general practitioner, ACR = American College of Rheumatology; IRGL = Invloed van Reuma op Gezondheid en Leefwijze (Influence of rheumatic diseases on health and lifestyle); VAS = visual analogue scale; WOMAC = Western Ontario and McMaster Universities Osteoarthritis Index; K–L = Kellgren and Lawrence score; PF = Physical Function subscale; NRS = numeric rating scale; HOOS = Hip disability and Osteoarthritis Outcome Score; OA = osteoarthritis. Italic script = time of measurements.

**Table 2 ijerph-17-07380-t002:** Risk of bias assessment.

Study	Random Sequence Generation	Allocation Concealment	Blinding of Participants and Personnel	Blinding of Outcome Assessment	Incomplete Outcome Data	Selective Reporting	Other Bias
V Baar 1998	+	+	-	-	+	?	+
Hopman-Rock 2000	?	?	-	-	?	?	+
Stener-Victorin 2004	+	?	-	-	-	+	+
Tak 2005	+	?	-	-	+	?	+
Fernandes 2010	+	+	-	-	+	+	+
Juhakoski 2011	+	+	-	-	+	?	+
French 2013	+	+	-	-	+	+	+
Abbott 2013	+	+	-	-	+	+	?
Villadsen 2014	+	+	-	-	+	+	+
Krauss 2014	+	+	-	-	+	+	+
Teirlinck 2016	+	+	-	-	+	+	+
Hermann 2016	+	+	-	-	+	+	?
Saw 2016	+	?	-	-	?	+	+
Bieler 2016	+	+	+	+	+	+	+

+ High risk of bias; - low risk of bias; ? unclear risk of bias.

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
