# Peer review of "Responders to Exercise Therapy in Patients with Osteoarthritis of the Hip: A Systematic Review and Meta-Analysis"

_ijerph, 2020, doi:10.3390/ijerph17207380_

Round 1

Reviewer 1 Report

Excellent study, few comments.

in abstract, clarify what OMERACT-OARSI is

in introduction, what is OARSI, clarify

In introduction, add the importance of OMERACT-OARSI, which makes a review and meta-analysis necessary

in table 1, clarify K-L (juhakoski 2011)

Author Response

Thank you for the opportunity to revise our manuscript. We also would like to thank the reviewers for their time, effort and well-thought comments.

Below you find our reply to each comment.

Reviewer 1:

Excellent study, few comments.

in abstract, clarify what OMERACT-OARSI is.
Reply: We added an extra sentence to the beginning of the abstract to explain this. The abstract counts 226 words now.

in introduction, what is OARSI, clarify
Reply: On page 2 (top)  we added a sentence to clarify this

In introduction, add the importance of OMERACT-OARSI, which makes a review and meta-analysis necessary
Reply: At the end of the introduction, we added a couple of sentences to better explain the importance of the OMERACT-OARSI responder criteria.

in table 1, clarify K-L (juhakoski 2011)
Reply: We added K-L to the abbreviations below table 1.

Reviewer 2 Report

Thank you for the opportunity to review “Responders to exercise therapy in patients with osteoarthritis of the hip: a systematic review and meta-analysis”. To be honest, I found this paper overwhelming, with simply small theory applications (introduction very poor scientifically) but not well organized, too many different kinds of previous studies and no central organizing themes, and no identified purpose or clear conclusions. The level of English writing is also problematic and frankly difficult to follow. In addition, the limitations addressed many obstacles for the scientific sounds of the systematic review as the authors established. 

Author Response

Thank you for the opportunity to revise our manuscript. We also would like to thank the reviewers for their time, effort and well-thought comments.

Below you find our reply to each comment.

Reviewer 2:

Thank you for the opportunity to review “Responders to exercise therapy in patients with osteoarthritis of the hip: a systematic review and meta-analysis”. To be honest, I found this paper overwhelming, with simply small theory applications (introduction very poor scientifically) but not well organized, too many different kinds of previous studies and no central organizing themes, and no identified purpose or clear conclusions. The level of English writing is also problematic and frankly difficult to follow. In addition, the limitations addressed many obstacles for the scientific sounds of the systematic review as the authors established.
Reply: Multiple changes were made. In the introduction we tried to better explain why we were interested in the OMERACT-OARSI criteria. We changed the conclusion, to make it better fit the aim mentioned in the introduction. We checked all spelling and sentences and made several changes. Furthermore, we had some difficulties understanding what was meant with too many different kind of previous studies and no central-organizing themes in our article in general. Therefore, if further changes are desirable, could you explain this a bit more? Thank you in advance.

Reviewer 3 Report

The main objective of this article was to fulfill Through a systematic review and meta-analysis the task from the National Health Care Institute of the Netherlands to find if exercise therapy was critical to the treatment for patients with hip osteoarthritis (OA). Through a systematic review and meta-analysis, the researchers found that exercise therapy was beneficial for patients with hip OA, although they also found that a large group of patients do not respond to exercise.

While in general the systematic review and meta-analysis follow solid statistical procedures, and the researchers were clear in stating the main goals of the study, some areas of improvement are:

1) It is unclear from methods of the systematic review; where was the systematic review registered? Prospero? There is no reference to the registration process, which is normally a very important in systematic reviews.

2) There is very solid statistical information on musculoskeletal disorders in the US via the Bone and Joint Burden (https://www.boneandjointburden.org/). Through the Global Alliance of for Musculoskeletal Health and/or data from the Ministry of Health of Netherlands, the authors should present some additional data pertinent to the paper: a) How many people are affected by hip OA, and more specifically how many are affected in the Netherlands to be needing this study? b) What is the economic impact of Hip OA on the patient and on the country? It would be interesting to briefly contrast this data with the global crisis.

3) As aforementioned while the methods are solid more information on the registration is needed.

4) Section 2.3 Risk of bias assessment was extremely well crafted, stating exactly how the researchers assessed bias. Section 2.1 Search Strategy would benefit being crafted in the same manner. The researchers should at the minimum add the search string used, which would save the reader time from having to then go the references to find what the previous reviewers used.

5) In Section 2.5 Outcome, for the ease of the reader the inclusion of a reference on the Likert scale should be included.

Minor edit required to fix: The flow chart and tables were easy to read and follow, however the font was not consistent between each chart, table, and paper.

Author Response

Thank you for the opportunity to revise our manuscript. We also would like to thank the reviewers for their time, effort and well-thought comments.

Below you find our reply to each comment.

Reviewer 3:

The main objective of this article was to fulfill Through a systematic review and meta-analysis the task from the National Health Care Institute of the Netherlands to find if exercise therapy was critical to the treatment for patients with hip osteoarthritis (OA). Through a systematic review and meta-analysis, the researchers found that exercise therapy was beneficial for patients with hip OA, although they also found that a large group of patients do not respond to exercise.

While in general the systematic review and meta-analysis follow solid statistical procedures, and the researchers were clear in stating the main goals of the study, some areas of improvement are:

1) It is unclear from methods of the systematic review; where was the systematic review registered? Prospero? There is no reference to the registration process, which is normally a very important in systematic reviews.
Reply: This systematic review started as a project in assignment of the National Health Care Institute of the Netherlands. A project plan/registration on forehand was presented and kept by the National Health Care Institute. The primary goal for our department was to establish a high-quality scientific report, which is published on the website of the National Health Care Institute. With this report the National Health Care Institute was able to formulate an advice to the Minister of Health, Welfare and Sports if exercise therapy for hip and knee OA should be reimbursed or not. Since the primary goal was not publication in scientific journals, we did not register our systematic review at PROSPERO, only at National Health Care Institute itself. We understand that this is not the normal process of systematic reviews, but the secondary analysis which we present in this article was an opportunity, and not planned upfront.

2) There is very solid statistical information on musculoskeletal disorders in the US via the Bone and Joint Burden (https://www.boneandjointburden.org/). Through the Global Alliance of for Musculoskeletal Health and/or data from the Ministry of Health of Netherlands, the authors should present some additional data pertinent to the paper: a) How many people are affected by hip OA, and more specifically how many are affected in the Netherlands to be needing this study? b) What is the economic impact of Hip OA on the patient and on the country? It would be interesting to briefly contrast this data with the global crisis.
Reply: We added information on prevalence and economic cost at the beginning of the introduction. We did not make an contrast to global data, because it was difficult to find the exact same data for a good comparison.

3) As aforementioned while the methods are solid more information on the registration is needed.
see our reply at 1.

4) Section 2.3 Risk of bias assessment was extremely well crafted, stating exactly how the researchers assessed bias. Section 2.1 Search Strategy would benefit being crafted in the same manner. The researchers should at the minimum add the search string used, which would save the reader time from having to then go the references to find what the previous reviewers used.
Reply: Since we used many search strings, we added the main search terms in the text and the full search to the appendix.

5) In Section 2.5 Outcome, for the ease of the reader the inclusion of a reference on the Likert scale should be included.
Reply: reference was included.

Minor edit required to fix: The flow chart and tables were easy to read and follow, however the font was not consistent between each chart, table, and paper.
Reply: we checked all fonts. Only the font in the flow chart and figures could not be altered/set to the same font as the paper, because there were created in a different program.

Round 2

Reviewer 2 Report

I would like to commend the authors on the revision of the manuscript and thank them for the point-by-point response. The paper is greatly improved. There are a couple of minor points that could do with further clarification.